# How Little Do We Know about HIV and STIs Prevention? Results from a Web-Based Survey among the General Population

**DOI:** 10.3390/healthcare10061059

**Published:** 2022-06-08

**Authors:** Andrea De Vito, Agnese Colpani, Beatrice Zauli, Maria Chiara Meloni, Marco Fois, Vito Fiore, Giovanni Antonio Pintus, Vincenzo Gesualdo Nardi, Sergio Babudieri, Giordano Madeddu

**Affiliations:** Unit of Infectious Diseases, Department of Medical, Surgery and Experimental Sciences, University of Sassari, 07100 Sassari, Italy; beatricezauli@gmail.com (B.Z.); m_chiara@hotmail.it (M.C.M.); mark.fois@hotmail.it (M.F.); vitofiore30010516@gmail.com (V.F.); anto.pintus@libero.it (G.A.P.); vi.nardi@tiscali.it (V.G.N.); babuder@uniss.it (S.B.); giordano@uniss.it (G.M.)

**Keywords:** HIV, STIs, sexual transmitted infections, U = U, education, prevention, survey

## Abstract

Background: Prevention campaigns have led to a significant decrease in new HIV diagnoses in Western Europe, while other sexual transmitted infections (STIs) have shown an opposite trend. Several educational programs are promoted among young students, whereas informational campaigns addressing the general population are scarce. We aimed to investigate the level of awareness regarding STIs among the general population. Methods: We proposed a questionnaire regarding STIs and HIV to the general population in Italy. We assigned 1 point to correct, 0.5 point to partially correct, and 0 point to wrong answers. We collected data about age, sex, region of origin, level of education and whether they were health workers. Results: Overall, 2183 people answered the questionnaire, of which 555 aged over 50 years old. Being male, older than 50 years old, retired or unemployed, not educated, and no regular use of condoms were associated with lower scores. Only 16% of participants knew the Undetectable = Untransmittable (U = U) campaign. Overall, 2131 (97.6%) people think more educational campaigns should be offered. Of interest, 80% said the questionnaire led them to learn more about HIV and STIs. Conclusion: Our study reveals several gaps in general population awareness about HIV and STIs, especially among people aged over 50 years old. Most participants stated that the questionnaire was a learning opportunity. These data suggest that improvement of knowledge could start from easy-to-dispose medium, such as surveys and questionnaires delivered through social media. Furthermore, particular attention should be paid to population segmentation and campaign tailoring to enhance interventions effectiveness. Our data reinforce the need for more informational and educational campaigns tailored to the specific segments of the population.

## 1. Introduction

Prevention campaigns have led to a significant decrease in new HIV diagnoses in West Europe [1,2], while other sexual transmitted infections (STIs) show an opposite trend, particularly syphilis and gonorrhea. In Italy, several educational programs are promoted among young students [3]. The United Nations Educational, Scientific and Cultural Organization (UNESCO) advocates a homogeneous educational agenda before 2030; in this regard, the International Technical Guidance on Sexuality Education has been published in 2009 and recently updated [4]. On this basis, the WHO published a document tailored on European countries [5]. In Italy, these guidelines are not yet applied, and national guidelines still need to be assessed; thus, the best way of delivering information and education and the optimal method to evaluate the outcome is not yet established [3].

Data about knowledge among students have been published, underlining widespread ignorance and risky behaviors [6,7,8,9,10]. These studies show precarious use of condoms with occasional partners. In addition, there are many gaps regarding the knowledge of method of transmission of HIV and STIs and proper way of prevention. Considering contraceptive pills, coitus interruptus and hygiene of genitals as effective preventive measures is a common mistake. Regarding the knowledge of STIs, educational campaigns have demonstrated their effectiveness [7,8,11,12]. However, it is more difficult to evaluate behavioral changes, since it would require years to evaluate an actual change in behavior. Data concerning the knowledge of STIs have also been obtained in specific populations at higher risk of contagion, such as sex workers, health workers and inmates [13,14,15,16]. On the contrary, fewer data about the knowledge of STIs among the general population are available; thus, it would be difficult to design targeted campaigns. In addition, there are no data about the knowledge of the “Undetectable = Untransmittable (U = U)” campaign, aiming to reduce HIV stigma, launched in February by the Prevention Access Campaign [17,18]. For these reasons, we aimed to investigate the level of awareness regarding STIs among the general population to outline the categories with a low knowledge level to target. 

## 2. Materials and Methods

We proposed an anonymous self-administered questionnaire regarding STIs and HIV to the general population in Italy. We promoted a web-based survey through google form, which was spread through social networks, asking associations and colleagues all over Italy to share it with their contacts.

The questionnaire included ten questions to investigate the level of awareness about HIV and STIs (Appendix A). It included multiple-choice questions; when only one correct response was required, we assigned 1 point for correct and 0 point for wrong answers. For multiple-choice questions with more than one correct answer, we assigned 1 point to correct, 0.5 point to partially correct, and 0 point to wrong answers.

The maximum score achievable was ten. In addition, we included questions about sexual behaviors, knowledge of U = U campaign, the perceived need for more educational campaigns and the perceived utility of the questionnaire. Moreover, we collected data about age, sex, region of origin, level of education and occupation. Regarding the level of education in Italy, the education system is organized in primary schools (6–10 years old), secondary school (11–13 years old), and high school (14–18 years old).

The research was conducted according to the Helsinki Declaration. The research does not contain clinical studies, and all participants’ data are fully anonymized. For this type of study, formal consent is not required according to current national law from Italian Medicines Agency, and according to the Italian Data Protection Authority, neither Ethical Committee approval nor informed consent was required for our study.

### 2.1. Sample Size

Assuming a conservative knowledge of 25% in the general population [19,20] and of 35% in the study population, based on an alpha error of 0.05 and beta error of 0.20, the estimated sample is equal to 658 subjects. Considering a missing response of 10%, the final sample size will be 724 subjects.

### 2.2. Statistical Analysis

Before performing the statistical analysis, data distribution was evaluated with the Kolmogorov–Smirnov test. Then, data were elaborated as numbers on total (percentages) and means ± standard deviation.

Continuous variables with parametric distribution were compared with Student’s t-test or with one-way ANOVA. Categorical variables were evaluated with the Pearson chi-squared test. The statistical significance level was established as *p* < 0.05.

The Stata statistical software package, version 16.1 (StataCorp LP, College Station, TX, USA) was used for data processing and statistical analysis.

## 3. Results

Overall, 2183 people answered the questionnaire, of which 555 (25.4%) were aged over 50 years old. The mean age was 39.8 ± 12.8 years; 1713 (78.5%) were female, and 457 (20.9%) were male. The main characteristics, with the mean (±SD) of the score, are summarized in Table 1.

Regarding the HIV and STIs methods of transmission, we reported poor knowledge (Figure 1); in particular, regarding the method of transmission of HIV, 130 (6.0%) people gave a wrong answer, 336 (15.4%) people gave a partially correct answer. Regarding the route of transmission of STIs, 526 (24.1%) people answered wrongly, whereas 97 (4.4%) gave a partially correct answer. The same was reported about the proper prevention of STIs and HIV, with 321 (14.7%) wrong answers. The most frequent mistake was considering genitalia sanitation after sexual intercourse a proper method of prevention.

Overall, the mean score achieved was 7.62 ± 1.42. Results from ANOVA analysis are reported in Table 2.

Regarding sex at birth, women answered better than men (*p* = 0.02), while sexual orientation was not associated with a statistically significant difference.

Regarding age, we compared mean scores from participants from decade to decade, finding a statistically significant gap. People over 60 years old obtained a lower mean score when compared to people aged 50–59, 40–49, 30–39, or 20–29 years old. In addition, people aged 50–59 years old performed worse than younger participants but better than those over 60 years old. Of notice, participants aged under 20 years old performed worse than participants aged 20–49 years; however, this difference was not statistically significant.

With regard to occupation, health workers performed better than anyone else, obtaining the highest mean score (8.42 ± 1.22), as expected. On the other hand, retired people had worse scores than anybody else. In addition, unemployed people showed poor knowledge; indeed, their mean score was lower than non-health-workers (*p* = 0.001). 

Moreover, we found a statistically significant difference in scores related to the level of education. Higher mean score was associated with higher level of education. The wider the gap in qualifications, the wider the gap in mean score. After performing the analysis without including health workers, the difference remained statistically significant.

The majority of people (96.6%) who answered the questionnaire declare having had at least one experience of sexual intercourse. Regarding behavioral data, 1611 (73.8%) people answered the question about using condoms; of these, 11.2% declared they never use condoms and 15.3% referred to occasional use. People over 50 years old have a high percentage of no use (15.2%) or sporadic use of condoms (16%). Meanwhile, 10.4% of people under 30 years old reported never using condoms, and 11.9% reported precarious use. In addition, the statistical analysis revealed that no use of condoms or precarious use of condoms was associated with a lower mean score.

We also investigated the perception of educational campaigns needed in Italy, and 2131 (97.6%) people think more educational campaigns should be offered. Interestingly, 80% of people said that the questionnaire has led them to learn more about HIV and STIs.

Finally, we investigated the level of awareness about the evidence that virologically suppressed people with HIV (PWH) cannot transmit the virus. Overall, 56.7% of the participants know that people with a not detectable HIV-RNA could not transmit the virus, but only 16.3% of participants knew about the U = U campaigns. In addition, 33.7% of participants answered that living with PWH is dangerous.

## 4. Discussion

Our study reveals several gaps in general population awareness about HIV and STIs. People aged over 50 years old and retired are the most uninformed. Since audience segmentation to design targeted educational and informational campaigns has shown its effectiveness [21], describing the most vulnerable populations is crucial. The same attention must be paid to younger individuals who have shown several gaps in knowledge of HIV and STIs. Many countries have already established national guidelines for school-based sex education. Unfortunately, Italy is not among them.

Regarding sex at birth, women gained higher scores. Of notice, 78.5% of participants were female at birth. These data suggest that women are better informed than men about HIV and STIs and are more prone to participate in this kind of intervention. 

As far as educational level is concerned, it is possible that people attending school are presented with educational campaigns more often. In addition, being more educated could guarantee a better understanding of the training received and therefore a better retention of information. 

The data about the use of condoms are particularly alarming when related to performance at the test; people who do not use condoms or use them discontinuously gained worse scores than others, suggesting that people who exposed themselves to the risk of infection are not aware of the risk they are exposed to. These data contrast with Trani et al. identifying little knowledge of STIs as a predictor for using condoms [6]. However, in our survey, we investigated the use of condoms during occasional sexual intercourses, while Trani and colleagues reported the use of condoms without including occasional or routine intercourses. In this regard, Trani and colleagues reported a higher perception of the risk as another predictor for condom use, the use of condoms was scarcer with usual partners. In addition, we found a higher percentage of no use or precarious use of condoms in older people, while younger people use condoms more than expected; the age-related trend is consistent with what was reported by Trani and colleagues [6]. However, young adults declared they use condoms “always” less often than what was reported by Trani and colleagues (51.8%) [6] but more than what was reported by Loconsole and colleagues (44.3% vs. 25.8%) [10]. These data reinforce the need for informational and educational campaigns to improve behavior and knowledge.

Our study has some limitations; firstly, we collected the answers in Italy, and we could not know if our results could be applied to other European countries. Secondly, since it was a web-based survey, the mean scores may be overestimated because participants could have looked up answers while filling in the questionnaire. However, we believe that people who voluntarily participated in an anonymous survey do not cheat. 

In addition, as this is a web-based survey, it could have ruled out people who are not familiar with social media and Internet or people who do not have access to the web. Moreover, people who decided to participate in the survey probably had higher awareness and interest in this topic, which could have led to a scoring overestimation.

Last but not least, we found a very low prevalence of participants knowing the U = U campaigns. These data are consistent with previous studies [12,22,23,24], even if we have few data about awareness among the general population. The U = U campaign has been promoted for a few years already, and it carries the important message that virologically suppressed PWH cannot transmit the virus [25,26]. This is important to fight stigma and self-stigma. In addition, talking about awareness among PWH revealed how important it is to increase serostatus disclosure and acceptance of antiretroviral therapy [18,24].

## 5. Conclusions

In conclusion, most participants stated that the questionnaire was a learning opportunity. These data suggest that improvement of knowledge could start from easy-to-dispose medium, such as surveys and questionnaires delivered through social media. Other studies, especially among students, reveal wide acceptance of interventions about sex education [11]. The real question is: which is the best way to deliver information and education? Regarding students, national guidelines on school-based sex education are mandatory. As far as elderly people are concerned, campaigns on HIV and STIs must be implemented locally and nationally. Particular attention should be paid to population segmentation and campaign tailoring to enhance interventions’ effectiveness.

## Figures and Tables

**Figure 1 healthcare-10-01059-f001:**
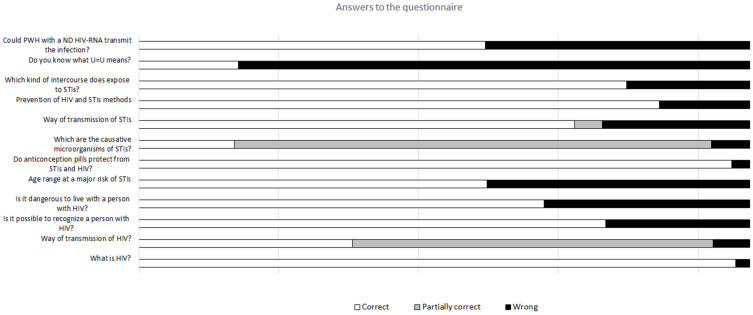
Answer to the 12 questions present in the survey.

**Table 1 healthcare-10-01059-t001:** Characteristics of 2183 patients that participated in the survey about HIV and STIs, and the mean (±SD) of the score.

Variables	*n*° of People	Mean	Standard Deviation	*p*-Value
Sex				0.0003
Female	1713	7.67	1.37	
Male	457	7.42	1.52	
Other *	13	6.73	2.79	
Region				0.0155
South	1281	7.56	1.48	
Center	222	7.84	1.21	
North	680	7.65	1.35	
Sexual Orientation				0.133
Heterosexual	1896	7.61	1.40	
Homosexual	95	7.86	1.43	
Bisexual	157	7.62	1.50	
Other **	10	8.1	0.69	
Did not answer	25	7.12	2.0	
Degree				<0.0001
Primary School	14	6.04	2.20	
Secondary School Diploma	203	6.92	1.45	
High School Diploma	905	7.51	1.39	
Academic Degree	1044	7.88	1.35	
Did not respond	17	6.86	1.38	
Profession				<0.0001
Unemployed	302	7.28	1.53	
Independent Work	291	7.61	1.27	
Dependent Work	996	7.62	1.33	
Student	226	7.52	1.41	
Health Worker	262	8.42	1.22	
Retiree	74	6.68	1.84	
Did not respond	32	7.09	1.61	
Age Range				<0.0001
<20	37	7.46	1.21	
20–29	544	7.69	1.50	
30–39	536	7.78	1.33	
40–49	511	7.73	1.29	
50–59	400	7.41	1.38	
>60	155	7.00	1.72	
Use of Condoms ***				<0.0001
Never	181	7.11	1.66	
Sometimes	247	7.41	1.52	
Often	357	7.83	1.32	
Always	667	7.78	1.36	
Not responded	159	7.31	1.49	

* Other: female transgender. male transgender; ** Asexual; Pansexual; Queer; *** only for people who have had sexual intercourse with occasional partner.

**Table 2 healthcare-10-01059-t002:** Questionnaire scores comparison of 2183 participants according to demographic, occupation, level of education, and sexual behaviors.

	Contrast	95%CI	*p*-Value
Sex			
Male vs. Female	−0.249	−0.424–0.075	0.002
Region			
Center vs. South	0.280	0.043–0.53	0.016
North vs. South	0.097	−0.060–0.254	0.318
North vs. Center	−0.187	−0.44–0.069	0.201
Sexual Orientation			
Homosexual vs. Heterosexual	0.249	−0.0157–0.655	0.450
Bisexual vs. Heterosexual	0.015	−0.305–0.336	1
Bisexual vs. Heterosexual	−0.233	−0.736–1.716	0.710
Degree			
Secondary vs. Primary School	0.883	−0.159–1.925	0.141
High School vs. Primary School	1.476	0.461–2.492	0.001
Academic Degree vs. Primary School	1.842	0.827–2.856	<0.001
High School vs. Secondary school	0.594	0.301–0.887	<0.001
Academic Degree vs. Secondary school	0.959	0.669–1.248	<0.001
Academic Degree vs. High school	0.365	0.193–0.536	<0.001
Profession			
Not health Worker vs. Unemployed	0.341	0.091–0.59	0.001
Student vs. Unemployed	0.243	−0.101–0.588	0.333
Health Workers vs. Unemployed	1.145	0.815–1.47	<0.001
Retired vs. Unemployed	−0.594	−1.102–0.087	0.011
Student vs. Not Health Worker	−0.097	−0.380–0.185	0.923
Health Workers vs. Not Health Worker	0.804	0.539–1.070	<0.001
Retired vs. Not Health Worker	−0.935	−1.403–0.467	<0.001
Health Workers vs. Student	0.902	0.547–1.257	<0.001
Retired vs. Student	−0.837	−1.362–0.313	<0.001
Retired vs. Health Worker	−1.739	−2.255–1.224	<0.001
Age Range			
20–29 vs. <20	0.226	−0.453–0.906	0.933
30–39 vs. <20	0.324	−0.356–10.004	0.751
40–49 vs. <20	0.269	−0.412–0.949	0.871
50–59 vs. <20	−0.048	−0.735–0.639	1.000
>60 vs. <20	−0.453	−10.185–0.279	0.488
30–39 vs. 20–29	0.098	−0.145–0.341	0.861
40–49 vs. 20–29	0.042	−0.204–0.289	0.997
50–59 vs. 20–29	−0.274	−0.538–0.011	0.035
>60 vs. 20–29	−0.679	−10.043–0.315	<0.001
40–49 vs. 30–39	−0.056	−0.303–0.191	0.988
50–59 vs. 30–39	−0.372	−0.637–0.108	0.001
>60 vs. 30–39	−0.777	−1.142–0.412	<0.001
50–59 vs. 40–49	−0.317	−0.584–0.050	0.009
>60 vs. 40–49	−0.722	−1.088–0.355	<0.001
>60 vs. 50–59	−0.405	−0.783–0.026	0.028
Use of Condoms			
Sometimes vs. Never	0.290	−0.083–0.666	0.212
Often vs. Never	0.713	0.364–1.062	<0.001
Always vs. Never	0.669	0.349–0.991	<0.001
Often vs. Sometimes	0.422	0.105–0.739	0.003
Always vs. Sometimes	0.379	0.093–0.664	0.003
Always vs. Often	−0.044	−0.295–0.208	0.990

## Data Availability

The data will be available upon specific request to the authors.

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
