# Peer review of "How Little Do We Know about HIV and STIs Prevention? Results from a Web-Based Survey among the General Population"

_healthcare, 2022, doi:10.3390/healthcare10061059_

Round 1
Reviewer 1 Report
Review report file attached

Author Response
Dear Reviewer,
We want to thank you for your careful reading and thoughtful comments on the previous draft.
We have carefully considered yourcomments in preparing our revision, which has resulted in a more precise, more compelling, and broader paper.
Please, find below our point-by-point response:
Reviewer (R): A brief summary A novel study that aimed to investigate the level of awareness regarding STIs among the general population in Italy. The contribution documented level of awareness on transmission and prevention of STIs and HIV in the country across demographic diversity including agegroups, sex and socio groups. The strength of the study included the use of web-based survey with a large sample size of over 2000 and methodology (self-administered, anonymous).
Author response (AR): Dear reviewer, we would like to thank you for having carefully read our manuscript and for giving us your precious comments.
General concept comments Review:
R: Specific comments Title Suggestion for improvement: Consider rephrasing title for clarity e.g., "How much do we know.... " about HIV and STIs? transmission ?prevention?
AR: Thank you for your comment. We choose to use “little” and not “much” to be provocative and to give the idea to the reader that there is much work to do about the information campaign. We added “prevention” in the title as suggested.
R: Please include reference for HIV (32). Chlamydia, one of the three STIs mentioned in the reference. May wish to include (36)
AR: Thank you for your comment. We add the references, as you suggested
R: Materials and methods (57 64): Well written methodology. Area for consideration: 1) How the survey responses were collected for anonymity 2) IRB approval 3) pretesting of the questionnaire.
AR: Thank you for your comment. The survey was conducted using google questionnaire, where people which decide to participate does not include any personal information that could let anyone to determine their identity. We better explain it in the method section. We have also added as supplemental material a translated version of the survey.
R: Consider mentioning any limitation of the study (after 175)
AR: We added a paragraph about the study limitations.
Reviewer 2 Report
This manuscript reports on a survey conducted in Italy of 2183 subjects to quantify their knowledge of sexually transmitted infections. Generally, more education
It seems the power analysis suggested a sample size of 724 subject, yet nearly 3 times that, (2183 subjects) were sampled. This seems to suggest that a more stringent p-value may be better suited to the data set collected.
Please expand upon the methods section so that the reader can tell how the survey was administered including the format (written, online, etc.) and how participants were selected. In answering this, it would be helpful to include a translated (if the survey was offered in Italian) version as an appendix. Additionally, the method of scoring the surveys needs to be addressed. If the survey had multiple choice answers, it could be straightforward to explain the scoring. If the survey had free-response answers, a rubric or guideline used for categorizing and scoring answers is needed.
Based on the heavily female general population, it seems as though the sampling may not have been random. Clarifying the sample selection in the methods will help here, but there should be discussion of how effective the random sampling method was and how the sample set relates to the general population.
Please ensure that an ethics compliance statement as required by the journal (see instructions for authors) is included in the manuscript accurately describing the author’s compliance with human subject research ethics.
Table 1 – It is unclear what is being reported with the mean and standard deviation in this table. I am assuming that this is the mean score and standard deviation on the 10-point STI knowledge assessment, but it is not clearly stated.
Minor comments
In parts of the world, secondary school and high school are used interchangeably. Please add discussion on the categorization of the degree variable to the introduction.
Figure 2 suggests there is a wrong answer to the question “Have you ever had a sexual intercourse? (sic).” Additionally, the grammar in figure 2 needs attention.
The abstract is very data heavy compared to a typical abstract and may be improved by focusing on conclusion from the data.
Line 35 – define the acronym UNESCO.
Line 62 – please provide a reference for the “Uundetectable=Untrasmittable (U=U) campaign (sic)” and correct the spelling. It would be better to give some background on this campaign in the introduction.
Author Response
Dear Reviewer:
We want to thank you for your careful reading and thoughtful comments on the previous draft.
We have carefully considered your comments in preparing our revision, which has resulted in a more precise, more compelling, and broader paper.
Please, find below our point-by-point response:
Reviewer (R): This manuscript reports on a survey conducted in Italy of 2183 subjects to quantify their knowledge of sexually transmitted infections. Generally, more education.
Author reply (AR): Dear reviewer, we would like to thank you for having carefully read our manuscript and giving us your precious comments, that helped us to improve the quality of the paper.
R: It seems the power analysis suggested a sample size of 724 subject, yet nearly 3 times that, (2183 subjects) were sampled. This seems to suggest that a more stringent p-value may be better suited to the data set collected.
AR: We performed the sample size calculation before starting to administer the questionnaire; also the statistical analysis was decided before the questionnaire administration; for this reason we decided to use the same statistical parameters that we settled in the study design.
R: Please expand upon the methods section so that the reader can tell how the survey was administered including the format (written, online, etc.) and how participants were selected. In answering this, it would be helpful to include a translated (if the survey was offered in Italian) version as an appendix. Additionally, the method of scoring the surveys needs to be addressed. If the survey had multiple choice answers, it could be straightforward to explain the scoring. If the survey had free-response answers, a rubric or guideline used for categorizing and scoring answers is needed.
AR: Thank you for your comment. We completely agree with you. We added a translated version of the questionnaire as supplemental materials and explained better the methods used.
R: Based on the heavily female general population, it seems as though the sampling may not have been random. Clarifying the sample selection in the methods will help here, but there should be discussion of how effective the random sampling method was and how the sample set relates to the general population.
AR Thank you for your answer. We believe that the reason for the presence of higher percentage of female is due to a higher concern and or interest about these topics by female compared to male. This could also explain why female had a higher score than men.
R: Please ensure that an ethics compliance statement as required by the journal (see instructions for authors) is included in the manuscript accurately describing the author’s compliance with human subject research ethics.
AR Thank you for your comment. We added a sentence in the method section.
R: Table 1 – It is unclear what is being reported with the mean and standard deviation in this table. I am assuming that this is the mean score and standard deviation on the 10-point STI knowledge assessment, but it is not clearly stated.
AR: We agree with you. We specify that it is the mean score to the survey.
R: Minor comments
In parts of the world, secondary school and high school are used interchangeably. Please add discussion on the categorization of the degree variable to the introduction.
AR: Thank you for your comment. In Italy the education system is organized in primary schools (6-10 years old), middle school (11-13 years old) and high school (14-18 years old). We added these definitions in the methods section of the manuscript.
R: Figure 2 suggests there is a wrong answer to the question “Have you ever had a sexual intercourse? (sic).” Additionally, the grammar in figure 2 needs attention.
AR: We agree with you. We removed this question from Figure 1, and added this information in the text.
R: The abstract is very data heavy compared to a typical abstract and may be improved by focusing on conclusion from the data.
AR: Thank you for your suggestion. We modified the abstract accordingly.
R: Line 35 – define the acronym UNESCO.
AR: Thank you for your suggestion. We provided it.
R: Line 62 – please provide a reference for the “Undetectable=Untrasmittable (U=U) campaign (sic)” and correct the spelling. It would be better to give some background on this campaign in the introduction.
AR: Thank you for your suggestion, we added a sentence regarding the U=U campaign in the introduction.
Round 2
Reviewer 2 Report
The authors have appropriately addressed my previous concerns in their revision. I have a few minor comments that remain:
Based on the author’s response, I still do not see the difference between secondary school diploma and high school diploma in table 1. Is the label for secondary school degree supposed to be middle school degree?
The grammar of the labels in figure 2 still needs attention.
It would be good to add to the new limitation discussion (lines 192-203) to mention the nature of the self-selected study population which is somewhat different than the general population.
Author Response
Dear author,
We would like to thank you for having re-read our paper and foryour thoughtful comments.
Please, find below our point-by-point response to reviewers (all revisions have been written in red):
Reviewer (R): The authors have appropriately addressed my previous concerns in their revision. I have a few minor comments that remain:
Based on the author’s response, I still do not see the difference between secondary school diploma and high school diploma in table 1. Is the label for secondary school degree supposed to be middle school degree?
Authors’ reply: Thank you for your comment. We clarify the definition of secondary school and high school in the methods section. In our country, mandatory schooling is divided into primary or elementary school (6-10 years old), secondary school (11-13 years old), and high school (14-18 years old).
R: The grammar of the labels in figure 2 still needs attention.
AR: we modified the Table 2 label as suggested.
It would be good to add to the new limitation discussion (lines 192-203) to mention the nature of the self-selected study population which is somewhat different than the general population.
AR: Thank you for your comment. We added the following sentence: Moreover, people who decided to participate in the survey probably had higher awareness and interest in this topic, which could have led to a scoring overestimation.